# Outcomes after perioperative SARS-CoV-2 infection in patients with proximal femoral fractures: an international cohort study

COVIDSurg Collaborative

**Correspondence to**
Mr Chetan khatri;
chetan.khatri@gmail.com

## ABSTRACT

**Objectives** Studies have demonstrated high rates of mortality in people with proximal femoral fracture and SARS-CoV-2, but there is limited published data on the factors that influence mortality for clinicians to make informed treatment decisions. This study aims to report the 30-day mortality associated with perioperative infection of patients undergoing surgery for proximal femoral fractures and to examine the factors that influence mortality in a multivariate analysis.

**Setting** Prospective, international, multicentre, observational cohort study.

**Participants** Patients undergoing any operation for a proximal femoral fracture from 1 February to 30 April 2020 and with perioperative SARS-CoV-2 infection (either 7 days prior or 30-day postoperative).

**Primary outcome** 30-day mortality. Multivariate modelling was performed to identify factors associated with 30-day mortality.

**Results** This study reports included 1063 patients from 174 hospitals in 19 countries. Overall 30-day mortality was 29.4% (313/1063). In an adjusted model, 30-day mortality was associated with male gender (OR 2.29, 95% CI 1.68 to 3.13, p<0.001), age >80 years (OR 1.60, 95% CI 1.1 to 2.31, p=0.013), preoperative diagnosis of dementia (OR 1.57, 95% CI 1.15 to 2.16, p=0.005), kidney disease (OR 1.73, 95% CI 1.18 to 2.55, p=0.005) and congestive heart failure (OR 1.62, 95% CI 1.06 to 2.48, p=0.025). Mortality at 30 days was lower in patients with a preoperative diagnosis of SARS-CoV-2 (OR 0.6, 95% CI 0.6 (0.42 to 0.85), p=0.004). There was no difference in mortality in patients with an increase to delay in surgery (p=0.220) or type of anaesthetic given (p=0.787).

**Conclusions** Patients undergoing surgery for a proximal femoral fracture with a perioperative infection of SARS-CoV-2 have a high rate of mortality. This study would support the need for providing these patients with individualised medical and anaesthetic care, including medical optimisation before theatre. Careful preoperative counselling is needed for those with a proximal femoral fracture and SARS-CoV-2, especially those in the highest risk groups.

**Trial registration number** NCT04323644

## BACKGROUND

The rapid worldwide spread of COVID-19, caused by the SARS-CoV-2 has had a severe effect on the elderly and frail population. A

## Strengths and limitations of this study

► This is a large, international, multicentre cohort study from which the results are generalisable across populations in other countries.
► This study described specific risk factors for mortality, which patients and those who care for them should use to make informed decisions regarding care.
► There is not control arm to assess contemporaneous patients with undergoing an operation for proximal femoral fractures without SARS-CoV-2 infection during the height of the pandemic. However with high-quality data present prepandemic strongly suggests a substantial increase in mortality.

fracture of the proximal femur (neck of femur fracture) is a critical event in the elderly, frail population, with a high rate of death despite medical and surgical intervention.[1] Since 2007, there has been a steady improvement in mortality after a proximal femoral fracture with 6.1% of patients dying within 30 days of injury in the UK in 2018.[2] However, the emergence of COVID-19 presents a new and unquantified risk to this particularly vulnerable group.

Proximal femur fractures represent a large international burden with incidence between 43 and 920 per 100000 population.[3] As most fractures of the proximal femur happen as a result of trips or falls in the home, people have continued to present with this injury despite social restrictions.[4 5] These patients typically have multiple comorbidities and frailty is common.[1] Resultantly, they are particularly vulnerable to pulmonary complications.[1 6] It is widely accepted that elderly patients with existing comorbidities are at higher risks of critical illness and mortality due to COVID-19, potentially due to a higher preponderance to release proinflammatory cytokines that result in severe disease.[7–9]

Clinicians have been swift to respond to this pandemic with large reorganisation of service provision.[10 11] In response to this, the COVIDSurg collaborative (www.globalsurg.org/covidsurg) has collected an international, large volume dataset to inform the global community of the safety of surgery in patients with perioperative SARS-CoV-2 infection. The first report has demonstrated a 30-day mortality of 23.8% across patients undergoing any type of surgery.[12] Data published so far have reported a high mortality rate in a small cohort of patients with proximal femoral fractures positive for SARS-CoV-2 infection, with a maximum cohort size of 114 patients (range 10–114 patients).[13–20] However, few reports have the sample size sufficient to explore the factors that influence outcome. Furthermore, large-scale data are required to explore preoperative and operative variables that influence outcomes in order to inform the clinical decision-making processes.

## Aims

The primary aim of this study is to determine the mortality rate observed in patients undergoing surgery for proximal femoral fracture with perioperative SARS-CoV-2 infection. Secondarily, we aim to explore the patient and treatment factors associated with these outcomes.

## METHODS

### Setting

This is an international, multicentre cohort study including consecutive patients who underwent surgery for proximal femoral fracture from 1 February 2020 to 3 April 2020. This study is a preplanned sub-analysis of a larger, ongoing study designed to assess outcomes following all surgery for patients with perioperative SARS-CoV-2 infection.[12]

The COVIDSurg collaborative is an international, multicentre, multidisciplinary team with individual collaborators collecting data locally, which is collated centrally. The collaborative methodology, which is well described and validated, was used for this project.[21]

### Inclusion criteria

Participating hospitals included consecutive patients undergoing surgery for proximal femoral fractures that had SARS-CoV-2 infection diagnosed (laboratory, clinical or radiologically) either 7 days preoperatively or up to 30 days postoperatively. For those diagnosed preoperative, this represents the timeframe where the majority of patients still active disease.[22] For those patients who underwent multiple procedures, the procedure closest to the time of confirmation of SARS-CoV-2 infection was defined as the index procedure.

Patients received laboratory confirmation of SARS-CoV-2 using quantitative reverse transcription Polymerase Chain Reseaction (qRT-PCR). As qRT-PCR is not available in all participating hospitals, patients were included if their diagnosis was made by clinical or radiological findings.

Clinical diagnosis was made in patients presenting with symptoms and a clinical pattern of COVID-19. These included cough, fever and/or myalgia.[23] Radiological diagnosis was made through CT scanning of the thorax according to local protocols. All patients who were included solely on clinical or radiological suspicion but had a subsequent negative qRT-PCR test were excluded from the database by individual collaborators.

### Diagnosis

This study includes all patients identified as having an operation for a proximal femoral fracture. The diagnosis was established pragmatically by the local site teams according to their assessment of the fracture. The reported data were screened by a central dedicated data cleaning team, with only confirmed proximal femoral fractures included in the cohort.

### Patient identification

Researchers at participating centres screened consecutive patients undergoing surgery to ensure all patients were identified. The study was initiated in some countries after their peak of infection, and therefore retrospective identification and data collection was permitted, as long as the data collection was consecutive at that site.

To reduce selection bias, a variety of written materials were distributed to site leads to highlight possible methods of identifying patients ensuring all eligible patients were included. Investigators were invited to social media groups and online teleconferences to troubleshoot recruitment issues, share learning and ensure consistent recruitment into the wider cohort.

### Outcome measures

The primary outcome measure was 30-day all-cause mortality, with the day of surgery defined as day zero. The secondary outcome measure was rate of pulmonary complications, which is a composite outcome defined previously from the Prevention of Respiratory Insufficiency after Surgical Management randomised controlled trial.[24 25]

Pulmonary complications were defined as pneumonia, acute respiratory distress syndrome and/or unexpected postoperative ventilation; these have been identified as the most frequent COVID-19 related pulmonary complications in medical patients.[23] Unexpected postoperative ventilation was defined as either: (1) any episode of non-invasive ventilation, invasive ventilation or extracorporeal membrane oxygenation after initial extubation following surgery or (2) unexpected failure to extubate following surgery.[12]

### Data collection and quality assurance

Data were collected online using the Research Electronic Data Capture web application.[26] Demographic variables recorded consisted of age, sex and American Society of Anesthesiologists (ASA) physical status classification. Age was collected as a categorical variable by deciles of age. ASA at the time of surgery was dichotomised to: (1)

grades 1–2 and (2) grades 3–5 for the purpose of analysis, time to surgery to (1) under 24 hours, (2) 24–48 hours and (3) over 48 hours and surgery to (1) hemiarthroplasty, (2) total hip replacement, (3) dynamic hip screw, (4) cannulated screws and (5) intramedullary nail. The timing of SARS-CoV-2 diagnosis was recorded as either preoperative or postoperative.

Before data were entered into analysis, site principle investigators were required to confirm all consecutive eligible cases had been completed and uploaded. Where diagnosis was unclear, authors were contacted for clarification.

## Statistical analysis

The study was reported according to Strengthening the Reporting of Observational Studies in Epidemiology guidelines.[27] Proportions are expressed with 95% CIs, and the mean and 95% CIs were used where data were assumed to be approximately normally distributed. Fisher's exact test was used for categorical data. Nonparametric data was summarised with the median and IQRs. Statistical significance was assessed at the 5% level.

The risk of death at 30 days was chosen as the primary outcome for the study. Mixed-effects logistic regression analysis was used to assess the strength and significance of associations between a number of explanatory variables and death within 30 days. Random effects were included in the mixed-effects model to account for the hierarchical structure of the data (individual hospital effects are naturally nested within country effects), and fixed effects were included to adjust for a range of preoperative variables that may influence mortality in this population and relevant factors related to the injury or treatment (eg, type of operation, time from admission to operation and type of anaesthetic). An additional analysis of the same factors was undertaken using the same model structure for the secondary outcome of pulmonary complications. This was an exploratory analysis with the significance level set at 5%, with no specific adjustments made for model testing. All analyses were implemented in R (R Core Team (2020). R: A language and environment for statistical computing. R Foundation for Statistical Computing, Vienna, Austria. URL https://www.R-project.org).

## Patient and public involvement

Patients were not involved in the design, conduct or reporting of this study.

## RESULTS
## Population

This study returned 30-day follow-up for 1063 patients with proximal femoral fractures. Data were collected in 174 hospitals from 19 countries (online supplemental table 1). Of these, 65.5% were female (696/1063). A percentage of 7.8% (83/1063) patients were <70 years old, 17.8% (189/1063) were between 70 and 79 years,

47.7% (507/1063) were between 80 and 89 years old and 26.7% (284/1063) were 90+ years old.

## Mortality

Overall 30-day mortality was 29.4% (313/1063). With each decile of age, mortality significantly increased, being highest in those patients >90 years old (38.7% (110/284), p=0.001).

In an adjusted model (figure 1), 30-day mortality was associated with male gender (OR 2.29, 95% CI 1.68 to 3.13, p<0.001), age >80 years (OR 1.60, 95% CI 1.1 to 2.31, p=0.013), diagnosis of dementia (OR 1.57, 95% CI 1.15 to 2.16, p=0.005), chronic kidney disease (OR 1.73, 95% CI 1.18 to 2.55, p=0.005) and congestive heart failure (OR 1.62, 95% CI 1.06 to 2.48, p=0.025). Thirty-day mortality was lower in patients with a preoperative diagnosis of SARS-CoV-2 (OR 0.60, 95% CI 0.42 to 0.85, p=0.004). Non-adjusted values are presented in online supplemental table 2.

## Pulmonary complications

In an adjusted model (figure 2), respiratory complications were associated with male gender (OR 1.7, 95% CI 1.27 to 2.28, p<0.001), diagnosis of dementia (OR 1.34, 95% CI 1.01 to 1.79, p=0.044) and congestive heart failure (OR 1.76, 95% CI 1.17 to 2.63, p=0.006). The presence of chronic obstructive pulmonary disorder dshowed no significant association (OR 1.42, 95% CI 0.96 to 2.09, p=0.076).

## Diagnosis

The majority of diagnosis of SARS-CoV-2 was made via PCR swab testing 93.3% (992/1063) (online supplemental tables 1 and 3), and there was no difference in mortality between those diagnosed clinically (p=0.668). The majority of patients 69% (733/1063) were diagnoses postoperatively.

## Preoperative variables

Preoperative symptoms (online supplemental table 4), including breathlessness, cough and fever (>38°C) were not significantly different in patients who were alive or dead at 30 days postoperatively. On examination of preoperative observations, a high respiratory rate was predictive of mortality (OR 1.73 95% CI 1.18 to 2.55, p=0.025) (figure 1). However, there was no significant difference in patient's heart rate, systolic or diastolic blood pressure (online supplemental table 5 and figure 1) between those who were alive or dead at 30 days.

Those patients with ASA grade 3–5 had a significantly higher mortality of 31.4% (281/899) versus ASA of 1–2 of 18.5% (28/151), p=0.001.

## Procedures

The operations were carried out under a general anaesthetic in 49.6% (527/1063) of patients (online supplemental table 6). A percentage of 67.2 (714/1063) of patients did not require any preoperative oxygen therapy. In this cohort, 31.8% (338/1063) of patients had their

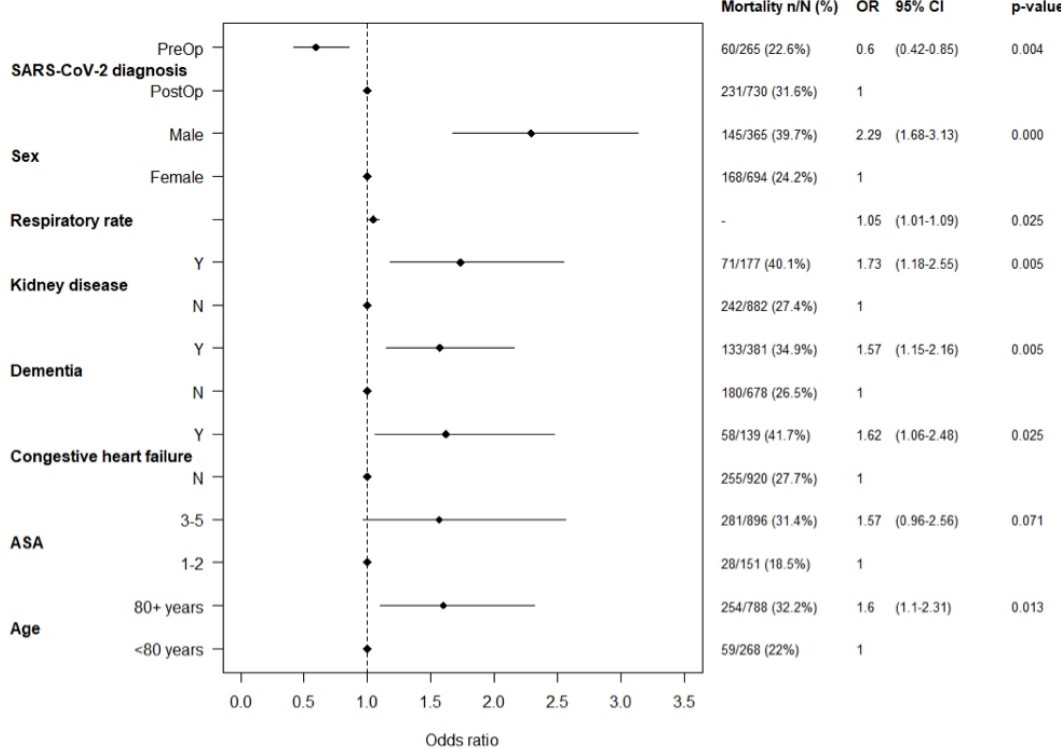

**Figure 1** Mixed-effects logistic regression model for 30-day mortality. ASA, American Society of Anesthesiologists.

operation within 24 hours of presentation to hospital, 21.1% (224/1063) had their operation between 24 and 47 hours and 19.2% (205/1063) of patients had their operation after 48 hours of presentation to hospital.

In this cohort, 45.1% (479/1063) of patients underwent haemiarthroplasty with a further 4.2% undergoing total hip replacement (45/1063). For patients who underwent fixation, 26% (276/1063) underwent dynamic hip screw fixation, 22.9% (243/1063) patients underwent intra-medullary fixation, 0.5% (5/1063) underwent cannu-lated screw fixation, while a further 1.4% (15/1053) underwent internal fixation.

There was no difference in mortality between patients undergoing general and regional anaesthesia (29.9% (157/527) vs 29.0% (152/524), p=0.787). However, there was an increased mortality in those patients requiring preoperative oxygen therapy (34.3% (115/336) vs 27.2% (194/714), p=0.031).

There was no significant difference in mortality for patients with delayed operation. The highest mortality was for patients operated between 24 and 47 hours of admission (34.4% (77/224)) but was not significantly higher than less than those operated after 48 hours (p=0.220).

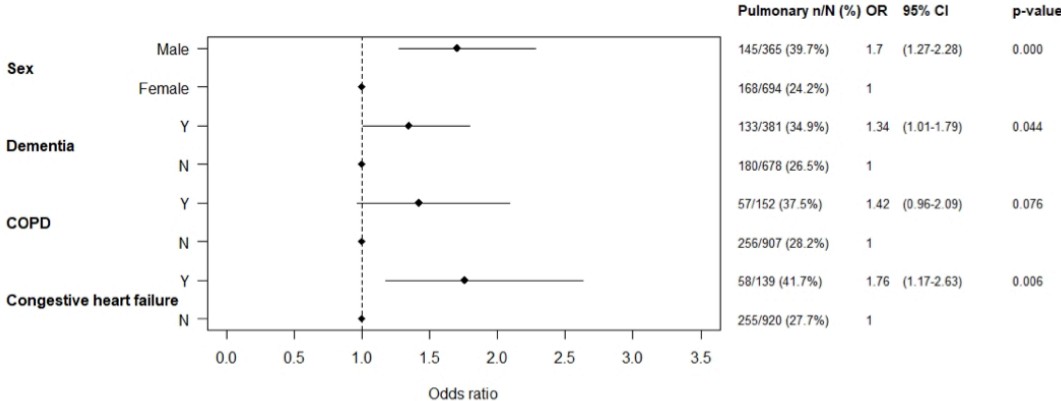

**Figure 2** Mixed-effects logistic regression model for pulmonary complications. COPD, chronic obstructive pulmonary disorder.

Mortality was highest in March (33.7%, 159/474) compared with April (27.0%, 150/558) and February (11.5%, 3/26), p=0.007 (online supplemental table 1).

## DISCUSSION

The 30-day mortality rate for patients with a perioperative diagnosis of SARS-CoV-2 infection undergoing surgery for proximal femoral fracture is substantial. An overall rate of 29.4% compares with the reported 30-day mortality in the literature for proximal femoral fractures ranging between 3.5% and 6.8%.[2 28–32] This rate is higher than found at the 1year time point.[33] Furthermore, elderly patients and those with medical comorbidities such as dementia, chronic kidney disease and congestive heart failure were associated with higher risk of 30-day mortality. Notably, patients with a preoperative diagnosis of SARS-CoV-2 infection had lower rates of 30-day mortality, likely reflecting early recognition and closer management of these patients. Findings from this study will be useful in guiding clinicians to identify high-risk patients that may warrant closer medical and surgical input during the COVID-19 pandemic.

Considering this high mortality, it is critical that patients who present without a diagnosis SARS-CoV-2 with proximal femoral fractures are protected from contracting SARS-CoV-2 in the perioperative period. A study by Kayani et al[17] has suggested that half of infections in patients with proximal femoral fractures occur in hospital, as denoted by having negative preoperative samples. Similarly, a study by Hall et al[34] has suggested nearly half of cases were due to nosocomial transmission. Within this study, 733 (69%) of infections were diagnosed postoperatively. This may infer that infections have been transferred in hospital, although due to incubation period of the virus, it is hard to know the proportion that contracted the virus prior to presentation or in hospital.[7 35] Higher mortality was observed in people who had a postoperative diagnosis, which emphasises the critical importance of avoiding in-hospital transmission. Hospitals should consider implementation of careful infection control processes to minimise and prevent transmission of SARS-CoV-2 infection. Within the elective setting, the creation of COVID-19 free surgical pathways for elective patients has been shown to reduce infection and subsequent mortality[36–38] and while only some of the principles are transferrable to the emergency setting, it demonstrates the value of meticulous infection control processes throughout the hospital stay. Furthermore, patients should be reinforced of methods to reduce risk of transmission in the community after discharge, including (but not limited to) social distancing, isolation and hygiene.

For those patients presenting with SARS-CoV-2 (either existing diagnosis or clinical findings suggestive of) and a proximal femoral fracture, it is important for data to be used as part of the informed consent process. In patients with multiple high-risk factors such as those who are more elderly, have respiratory and cardiac comorbidities,

non-operative management may be considered following an appropriate discussion with the patient and/or their family. Every year in the UK, 2.5% of hip fractures are treated non-operatively.[39] A study performed before the pandemic reported that the mortality within 30 days for conservatively treated patients was 31.3%.[40] We do not know the mortality from non-operative management during the pandemic for patients with SARS-Cov-2, but the particularly high mortality associated with surgery in high-risk groups may change the balance of benefit and harm towards conservative treatment, and this should be considered.

The 30-day mortality of 29.4% identified within this study is comparable with published literature in the UK (range from 16.3% to 35.6%),[15–17 19 34 41] Italy (18.75%),[14] Spain (30.4%)[13] and the USA (range from 35.3% to 56%).[18 20] From a study within the UK, the authors also found a correlation between male sex and increased mortality (OR 2.69), which is similar to that demonstrated in this study (OR 2.29).[16] Additionally, another UK study reported having more than three comorbidities as a risk factor for mortality.[17] This study has specifically delineated a diagnosis of dementia, chronic kidney disease and congestive heart failure as being independent risk factors for mortality. In a study from USA, the authors found those patients who died were older with multiple comorbidities, and this was reflected in statistically significant higher ASA scores in comparison with their negative counterparts.[20]

This study found that there was no significant increase in mortality with delay to surgery. Current guidelines suggest early surgery should be undertaken,[42] and this is associated with lower mortality.[43] This would suggest that those patients at the highest risk of mortality can have medical optimisation, if appropriate, and will not result in a higher mortality from SARS-CoV-2 infection. This includes correction of concurrent medical issues often found in this population, examples of which include correction of acute renal failure, electrolyte disturbances and/or anticoagulation related issues. With regards to recovery from SARS-CoV-2 infection, it is important to consider that an increased risk of mortality for those undergoing surgery persists until 7weeks after diagnosis.[44] This risk reduces gradually after 2weeks after diagnosis and should be considered.

Similarly, previous studies have found a higher rate of mortality in patients undergoing general versus regional anaesthesia for proximal femoral fractures.[45 46] This study reports no difference between general and regional anaesthetic (29.9% vs 29.0%, p=0.787). While this was not the primary outcome of this study, this suggests that a positive test for SARS-CoV-2 should not have a large influence on anaesthetic decisions. This should be interpreted with caution in the light of this being an exploratory study. Out of all clinical features, respiratory rate at presentation was associated with higher mortality. Clinicians should focus on this as an important finding when counselling patients of their perioperative mortality.

This study has also found an increased mortality during the month of March 2020. This corresponds to the peak of caseload of infections internationally.[47][48] Increased circulation of SARS-CoV-2 within countries has shown to increase mortality through higher viral loads.[47][49] This study validates that surgical patients are particularly susceptible during surge of cases.

This is a large, varied cohort of patients undergoing surgery for a proximal femoral fractures with SARS-CoV-2 infection diagnosed perioperatively. This study was conducted in multiple centres, internationally, allowing it to be generalisable across populations in other countries.

## Strength and Limitations

This study was conducted in hospitals in the early to midphase of the pandemic where routine testing was not available in all participating centres. As such, to be pragmatic, patients were included if a clinical diagnosis was made by the treating physician. Protocols were not standardised for clinical diagnosis and were left the senior treating physician. Laboratory diagnosis was made by qRT-PCR, from which false-negative results may have excluded patients from analysis. Indeed, the sensitivity of qRT-PCR testing for has shown to be as low as 32% for throat swabs.[50] However, in patients with negative results and high clinical suspicion of SARS-CoV-2 infection, multiple samples are often taken, including broncho-alveolar lavage. Thus, the number of patients excluded is expected to be low. While this study reports a higher mortality from postoperative diagnosis of SARS-CoV-2 infection, it is unclear whether the infection was contracted preoperatively or not, as has been discussed previously.

This study does not have a control arm, assessing contemporaneous patients with undergoing an operation for proximal femoral fractures without SARS-CoV-2 infection during the height of the pandemic. However, comparison with high-quality prepandemic data strongly suggests a substantial increase in mortality. Patients and those who care for them should consider this carefully when making decisions in this common and challenging clinical scenario.

## CONCLUSION

Patients undergoing surgery for a proximal femoral fracture with a peri-operative infection of SARS-CoV-2 have a high rate of mortality. The study would support the approach of providing these patients with individualised medical and anaesthetic care, including medical optimisation before theatre. It is imperative to prevent transmission of COVID-19 in the hospital setting. Careful preoperative counselling is needed for those with a proximal femoral fracture and SARS-CoV-2, especially those in the highest risk groups.

**Acknowledgements** Xiao Liu and Alistair Denniston, Royal College of Surgeons of England COVID-19 Research Group.

**Collaborators** COVIDSurg authorship list: Writing group: C Khatri, A E Ward, D Nepogodiev, I Ahmed, D Chaudhry, F Dhaif, B Bankhad-Kendall, H Kaafarani, C Bretherton, A Mahmood, L Marais, N Parsons, A Bhangu, A Metcalfe (senior author & overall guarantor). Statistical analysis: N Parsons (lead statistician), C Khatri. Operations Committee: K Siaw-Acheampong, D Chaudhry, B E Dawson, J P Evans, J C Glasbey, R R Gujjuri, E Heritage, C S Jones, S K Kamarajah, C Khatri, J M Keatley, S Lawday, E Li, S C Mckay, D Nepogodiev, G Pellino, A Tiwari, J F F Simoes, I M Trout, M L Venn, R J W Wilkin, A Bhangu. Dissemination Committee: A O Ademuyiwa, A Agarwal, E Al Ameer, D Alderson, O Alser, A P Arnaud, K M Augestad, B Bankhead-Kendall, R A Benson, S Chakrabortee, R Blanco-Colino, A Brar, A Minaya Bravo, K A Breen, I Lima Buarque, E Caruana, M F Cunha, G H Davidson, A Desai, S Di Saverio, J Edwards, M Elhadi, S Farik, M Fiore, J E Fitzgerald, S Ford, G Gallo, D Ghosh, G M A Gomes, E Griffiths, C Halkias, E M Harrison, P Hutchinson, A Isik, H Kaafarani, A Kolias, I Lawani, H Lederhuber, S Leventoglu, M W Loffler, J Martin, H Mashbari, D Mazingi, H Mohan, R Moore, D Moszkowicz, J S Ng-Kamstra, S Metallidis, S Moug, M Niquen, F Ntirenganya, O Outani, F Pata, T D Pinkney, P Pockney, D Radenkovic, A Ramos-De la Medina, K Roberts, I Santos, A Schache, A Schnitzbauer, G D Stewart, R Shaw, S Shu, K Soreide, A Spinelli, S Sundar, S Tabiri, P Townend, G Tsoulfas, G van Ramshorst, R Vidya, D Vimalachandran, N Wright, EuroSurg, European Society of Coloproctology, GlobalSurg, GlobalPaedSurg, ITSurg, PTSurg, SpainSurg, S-ECCO, J F F Simoes (chair). Collaborators (*asterix indicates principle investigator): Mak JKC, Kulkarni R, Sharma N, Nankivell P, Tirotta F, Parente A, Breik O, Kisiel A, Cato LD, Saeed S, Bhangu A*, Griffiths E*, Pathanki AM, Ford S*, Desai A*, Almond M*, Kamal M (Queen Elizabeth Hospital Birmingham, United Kingdom); Chebaro A*, Lecolle K, Truant S, El Amrani M, Zerbib P, Pruvot FR, Mathieu D, Surmei E, Mattei L, Marin H (CHU Lille, France); Dudek J, Singhal T*, El-Hasani S (Princess Royal University Hospital, United Kingdom); Nehra D*, Walters A, Cuschieri J, Davidson GH (Harborview Medical Center, United States); Ho M*, Wade RG*, Johnstone J, Bourke G, Brunelli A, Elkadi H, Otify M*, Pompili C, Burke JR, Bagouri E, Chowdhury M, Abual-Rub Z, Kaufmann A, Munot S, Lo T*, Young A, Kowal M, Wall J, Peckham-Cooper A (The Leeds Teaching Hospitals NHS Trust, United Kingdom); Winter SC*, Belcher E*, Stavroulias D, Di Chiara F, Wallwork K, Qureishi A, Lami M, Sravanam S, Mastoridis S, Shah K, Chidambaram S, Smillie R, Shaw AV, Bandyopadhyay S, Cernei C, Bretherton C, Jeyaretna D*, Ganau M*, Piper RJ*, Duck E, Brown S, Jelley C, Tucker SC*, Bond-Smith G*, Griffin XL*, Tebala GD, Neal N*, Vatish M*, Noton TM, Ghattaura H, Maher M, Fu H, Risk OBF, Soleymani majd H, Sinha S*, Shankar S, Aggarwal A, Kharkar H, Lakhoo K*, Verberne C, Mastoridis S (Oxford University NHS Foundation Trust, United Kingdom); Senent-Boza A*, Sánchez-Arteaga A, Benítez-Linero I, Manresa-Manresa F, Tallón-Aguilar L, Melero-Cortés L, Fernández-Marín MR, Durán-Muñoz-Cruzado VM, Ramallo-Solís I, Beltrán-Miranda P, Pareja-Ciuró F, Antón-Eguía BT (Hospital Universitario Virgen Del Rocío, Spain); Dawson AC*, Drane A (Gosford Hospital, Australia); Oliva Mompean F, Gomez-Rosado J*, Reguera-Rosal J, Valdes-Hernandez J, Capitan-Morales L, del Toro Lopez MD (Hospital Universitario Virgen Macarena, Spain); Patel M, Shabana A, Alanbuki A*, Usman O (Brighton and Sussex NHS Trust, Princess Royal Hospital, United Kingdom); Tang A, Beamish AJ, Price C, Bosanquet D*, Magowan D, Solari F, Williams G, Nassa H, Smith L (Royal Gwent Hospital, United Kingdom); Elliott L, Mccabe G, Holroyd D*, Jamieson NB (Glasgow Royal Infirmary, United Kingdom); Mariani NM*, Nicastro V (Asst Santi Paolo E Carlo, Italy); Li Z*, Parkins K, Spencer N, Harries R, Egan RJ, Motter D, Jenvey C, Mahoney R, Fine N, Minto T, Henry A (Morriston Hospital Swansea, United Kingdom); Gill C, Dunne N, Sarma DR, Godbole C, Carlos W, Tewari N*, Jeevan D, Naredla P, Khajuria A, Connolly H, Robertson S, Sweeney C, Di Taranto G, Shanbhag S, Dickson K, McEvoy K, Skillman J, Sait M, Al-omishy H, Baig M, Heer B (University Hospitals Coventry and Warwickshire NHS Trusts, United Kingdom); Lunevicius R*, Sheel ARG, Sundhu M, Santini AJA, Fathelbab MSAT, Hussein KMA, Nunes QM, Jones RP, Shahzad K, Haq I, Baig MMAS, Hughes JL, Kattakayam A, Rajput K, Misra N, Shah SB, Clynch AL, Georgopoulou N, Sharples HM, Apampa AA, Nzenwa IC, Sud A (Aintree Hospital, Liverpool University Hospitals NHS Foundation Trust, United Kingdom); Podolsky D*, Coleman NL, Callahan MP (NewYork-Presbyterian / Columbia University Medical Center, United States); Dunstan M*, Beak P, Gerogiannis I* (Kingston Hospital NHS Foundation Trust, United Kingdom); Ebrahim A, Alwadiya A, Goyal A*, Phillips A, Bhalla A*, Demetriou C, Grimley E, Theophilidou E, Ogden E, Malcolm FL, Davies-Jones G, Ng JCK, Mirza M, Hassan M, Elmaleh N, Daliya P, Williams S, Bateman A*, Chia Z (Royal Derby Hospital, United Kingdom); A'Court J, Konarski A, Faulkner G* (Royal Bolton Hospital, United Kingdom); Talwar R*, Patel K, Askari A, Jambulingam P S, Shaw S, Maity A, Hatzantonis C, Sagar J, Kudchadkar S, Cirocchi N, Chan CH (Luton and Dunstable University Hospital, United Kingdom); Eberbach H*, Bayer J*, Erdle B, Sandkamp R (Medical Center – Albert-Ludwigs-University of Freiburg, Faculty of Medicine, Germany); Kaafarani H*, Breen K, Bankhead-Kendall B, Alser O, Mashbari H, Velmahos G, Maurer LR, El Moheb M, Gaitanidis A, Naar L, Christensen MA, Kapoen C, Langeveld K, El Hechi M, Mokhtari A (Massachusetts General Hospital, United States); Main B, Maccabe T, Newton C*, Blencowe NS, Fudulu DP, Bhojwani D, Baquedano M, Caputo M*, Rapetto F, Flannery, O, Hassan A (University Hospitals

Bristol and Weston NHS Foundation Trust, United Kingdom); Edwards J*, Ward A, Tadross D, Majkowski L, Blundell C, Forlani S, Nair R, Guha S, Brown SR, Steele C, Kelty CJ*, Newman T, Lee M, Chetty G*, Lye G, Balasubramanian SP, Sureshkumar Shah N, Sherif M, Al-mukhtar A, Whitehall E, Giblin A, Wells F, Sharkey A, Adamec A, Madan S (Sheffield Teaching Hospital NHS Foundation Trust, United Kingdom); Konsten J*, Van Heinsbergen M (VieCuri Medisch Centrum, Netherlands); Sou A, Simpson D, Hamilton E*, Blair J (Shrewsbury and Telford Hospitals, United Kingdom); Jimeno Fraile J*, Morales-Garcia D, Carrillo-Rivas M, Toledo Martínez E, Pascual À (Marqués De Valdecilla University Hospital, Spain); Landaluce-olavarria A*, Gonzalez De miguel M, Fernández Gómez Cruzado L, Begoña E, Lecumberri D (Hospital Urduliz, Spain); Calvo Rey A, Prada hervella GM*, Dos Santos Carregal L, Rodriguez Fernandez MI, Freijeiro M, El Drubi Vega S (Hospital Clinico Universitario De Santiago De Compostela (Chus), Spain); Van den Eynde J, Oosterlinck W*, Van den Eynde R, Sermon A, Boeckxstaens A, Cordonnier A, De Coster J, Jaekers J, Politis C, Miserez M (UZ Leuven, Belgium); Galipienso Eri M, Garcia Montesino JD, Dellonder Frigolé J, Noriego Muñoz D* (Hospital Universitari De Girona Dr. Josep Trueta, Spain); Lizzi V*, Vovola F, Arminio A, Cotoia A, Sarni AL (Ospedali Riuniti Azienda Ospedaliera Universitaria Foggia, Italy); Bekheit M*, Kamera BS, Elhusseini M, Sharma P, Ahmeidat A, Gradinariu G, Cymes W, Hannah A, Mignot G, Shaikh S*, Agilinko J (Aberdeen Royal Infirmary, United Kingdom); Sgrò A, Rashid MM, Milne K, McIntyre J, Akhtar MA*, Turnbull A, Brunt A, Stewart KE (Victoria Hospital Kirkcaldy, United Kingdom); Wilson MSJ*, Rutherford D, McGivern K, Massie E (Forth Valley Royal Hospital, United Kingdom); Duff S*, Moura F, Brown BC, Khan A, Asaad P, Wadham B, Aneke IA, Collis J, Warburton H (Wythenshawe Hospital, Manchester University NHS Foundation Trust, United Kingdom); Thomas M, Pearce L*, Fountain DM, Laurente R, Sigamoney KV, Dasa M, George K*, Naqui Z*, Galhoum M, Lipede C, Gabr A, Radhakrishnan A, Hasan MT, Kalenderov R, Pathmanaban O, Colombo F, Chelva R (Salford Royal NHS Foundation Trust, United Kingdom); Subba K, Abou-Foul AK, Khalefa M, Hossain F*, Moores T, Pickering L (Walsall Manor Hospital NHS Trust, United Kingdom); Shah J*, Anthoney J, Emmerson O (North Manchester General Hospital, United Kingdom); Bevan K*, Makin-Taylor R, Ong CS, Callan R, Bloom O (Bedford Hospital, United Kingdom); Vidya R*, Chauhan G, Kaur J, Burahee A, Bleibleh S, Pigadas N*, Snee D, Bhasin S*, Crichton A, Habeebullah A, Bodla AS, Yassin N*, Mondragon M, Dewan V (Royal Wolverhampton NHS Trust, United Kingdom); Giuffrida MC*, Marano A, Palagi S, Di Maria Grimaldi S, Testa V, Peluso C, Borghi F, Simonato A, Puppo A, D'Agruma M, Chiarpenello R, Pellegrino L, Maione F, Cianflocca D, Pruiti Ciarello V, Giraudo G, Gelarda E, Dalmasso E, Abrate A, Daniele A, Ciriello V, Rosato F, Garnero A, Leotta L (Santa Croce E Carle Hospital, Cuneo, Italy); Chiozza M, Anania G*, Urbani A, Koleva Radica M, Carcoforo P*, Portinari M, Sibilla M (Azienda Ospedaliero Universitaria Sant'Anna, Italy); Archer JE*, Odeh A, Siddaiah N (Royal Orthopaedic Hospital, United Kingdom); Baumber R, Parry J* (Royal National Orthopaedic Hospital, United Kingdom); Carmichael H, Velopulos CG*, Wright FL, Urban S, McIntyre Jr RC, Schroeppel TJ*, Hennessy EA, Dunn J*, Zier L (University of Colorado Hospital/Memorial Hospital/Medical Center of the Rockies, United States); Parmar C*, Mccluney S, Shah S (The Whittington Hospital, United Kingdom); Muñoz Vives JM, Osorio A, Gómez Díaz CJ, Guariglia CA, Soto Montesinos C*, Sanchon L, Xicola Martínez M, Guàrdia N, Collera P, Diaz Del Gobbo R, Sanchez Jimenez R, Farre Font R, Flores Clotet R (Fundació Althaia - Xarxa Assistencial Universitària de Manresa, Spain); Brathwaite CEM*, Liu H, Petrone P, Hakmi H, Sohail AH, Baltazar G, Heckburn R (NYU Langone Health-NYU Winthrop Hospital, United States); Aujayeb A, Townshend D*, McLarty N, Shenfine A, Jackson K, Johnson C (Northumbria NHS Hospital Trust, United Kingdom); Madhvani K, Hampton M, Hormis AP* (Rotherham District General Hospital, United Kingdom); Young R*, Miu V, Sheridan K, MacDonald L, Green S, Onos L (York Teaching Hospitals NHS Trust, United Kingdom); Dean B*, Luney C, Myatt R, Williams MA, McVeigh J (Nuffield Orthopaedic Centre, United Kingdom); Alqallaf A, Ben-Sassi A*, Mohamed I, Mellor K, Joshi P, Joshi Y (Wrexham Maelor Hospital, United Kingdom); Crichton R*, Sonksen J, Aldridge K (Russell's Hall Hospital, Dudley, United Kingdom); Layton GR*, Karki B, Jeong H, Pankhania S, Asher S, Folorunso A, Mistry S, Singh B, Winyard J, Mangwani J (Leicester General Hospital, United Kingdom); Babu BHB*, Liyanage ASD, Newman S, Blake I, Weerasinghe C (Southport and Formby District General Hospital, United Kingdom); Ballabio M*, Bisagni P, Longhi M, Armao T, Madonini M, Gagliano A, Pizzini P (Ospedale Maggiore Di Lodi, Italy); Älgå A*, Nordberg M, Sandblom G (South General Hospital, Sweden); Jallad S, Lord J, Anderson C, El Kafsi J*, Logishetty K*, Saadya A, Midha R, Ip M, Subbiah Ponniah H, Stockdale T, Bacarese-Hamilton T, Foster L, James A, Anjarwalla N, Marujo Henriques D, Hettige R, Baban C, Tenovici A, Salerno G (Wexham Park, Frimley Health NHS FT, United Kingdom); Hardie J, Page S, Anazor F, King SD, Luck J, Kazzaz S* (Frimley Park, Frimley Health NHS FT, United Kingdom); HKruijff S*, De Vries JPPM, Steinkamp PJ, Jonker PKC, Van der Plas WY, Bierman W, Janssen Y (University Medical Center Groningen, Netherlands); Borgstein ABJ, Gisbertz SS*, van Berge Henegouwen MI* (Amsterdam UMC VUMV, Netherlands); Enjuto D*, Perez Gonzalez M, Díaz Peña P,

Gonzalez J, Marqueta De Salas M, Martinez Pascual P, Rodríguez Gómez L, Garcés García R, Ramos Bonilla A, Herrera-Merino N, Fernández Bernabé P, Cagigal Ortega EP, Hernández I, García de Castro Rubio E, Cervera I (Severo Ochoa University Hospital, Spain); Kashora F*, Siddique MH, Singh A, Barmpagianni C, Basgaran A, Basha A, Okechukwu V, Bartsch A, Gallagher P, Maqsood A, Sahnan K, Leo CA, Lewis SE, Ubhi HK, Exley R, Khan U, Shah P, Saxena S, Zafar N, Abdul-Jabar H (Northwick Park Hospital, United Kingdom); Mongelli F, Bernasconi M, Di Giuseppe M, Christoforidis D*, La Regina D, Arigoni M (Ente Ospedaliero Cantonale, Switzerland); Liew I*, Al-Sukaini A, Mediratta S, Saxena D (Queen Elizabeth Hospital King's Lynn, United Kingdom); Brown O, Boal M, Dean H, Higgs S*, Stanger S, Abdalaziz H, Constable J, Ishii H, Preece R, Dovell G, Gopi reddy R (Gloucestershire Royal Hospital, United Kingdom); Dehal A* (Kaiser Permanente Panorama City Medical Center, United States); Shah HB, Cross GWV, Seyed-Safi P*, Smart YW, Kuc A, Al-Yaseen M (Watford General Hospital, United Kingdom); Jayasankar B, Balasubramaniam D*, Abdelsaid K, Mundkur N, Gallagher B* (Tunbridge Wells Hospital, United Kingdom); Hine T, Keeler B*, Soulsby RE, Taylor A* (Milton Keynes University Hospital, United Kingdom); Davies E*, Ryska O, Raymond T, Rogers S*, Tong A, Hawkin P (Royal Lancaster Infirmary, United Kingdom); Kinnaman G, Meagher A*, Sharma I, Holler E (Iu Health Methodist Hospital, United States); Dunning J, Viswanath Y, Freystaetter K, Dixon J*, Hadfield JN, Hilley A, Egglestone A, Smith B (James Cook University Hospital, United Kingdom); Arkani S*, Freedman J* (Danderyds Hospital, Sweden); Youssef M*, Sreedharan L, Baskaran D, Shaikh I, Seebah K, Reid J, Watts D, Kouritas V, Chrastek D, Maryan G, Gill DF, Khatun F (Norfolk and Norwich University Hospital, United Kingdom); Ranjit S, Parakh J, Sarodaya V, Daadipour A, Khalifa M (Newham University Hospital, United Kingdom); Bosch KD*, Bashkirova V, Dvorkin LS, Kalidindi VK (North Middlesex University Hospital, London, United Kingdom); Choudhry A, Marx W* (Suny Upstate University Hospital, United States); Espino Segura-Illa M, Sánchez Aniceto G*, Castaño-Leon AM*, Jimenez-Roldan L, Delgado Fernandez J, Pérez Núñez A, Lagares A, Garcia Perez D, Santas M, Paredes I, Esteban Sinovas O, Moreno-Gomez L, Rubio E*, Vega V, Vivas Lopez A, Labalde Martinez M, García Villar O, Pelaéz Torres PM, Garcia - Borda J, Ferrero Herrero E, Gomez P, Eiriz Fernandez C, Ojeda-Thies C*, Pardo Garcia JM (12 De Octubre University Hospital, Spain); Wynn Jones H, Divecha H*, Whelton C, Board T (Wrightington, Wigan & Leigh NHS Foundation Trust, United Kingdom); Hardie C, Powell-Smith E* (Harrogate District Hospital, United Kingdom); Alotaibi M*, Maashi A, Zowgar A, Alsakkaf M (King Faisal Hospital, Saudi Arabia); Izquierdo O, Ventura D, Castellanos J (Parc Sanitari Sant Joan de Déu, Sant Boi de Llobregat, Barcelona, Spain); Lara A, Escobar D, Arrieta M, Garcia de cortazar U*, Villamor Garcia I (Hospital Universitario De Basurto, Spain); Cioci A, Ruiz G*, Allen M, Rakoczy K, Pavlis W, Saberi R (University of Miami Hospital, United States); Sobti A, Khaleel A*, Unnithan A, Memon K, Pala Bhaskar RR, Maqboul F, Kamel F, Al-Samaraee A, Madani R*, Kumar L, Nisar P, Agrawal S* (Ashford and St Peter's Hospital, United Kingdom); Llaquet Bayo H* (Hospital de Palamós-SSIBE, Spain); Duchateau N, De Gheldere C* (Heilig Hart Ziekenhuis Lier, Belgium); Martin J*, Cheng D, Yang H, Fayad A (London Health Sciences Centre and St Josephs Health Care London, Canada); Wood ML, Persad A, Groot G*, Pham H (Saskatoon City Hospital/Royal University Hospital/St. Paul's Hospital, Canada); Hakami I*, Boeker C, Mall J* (KRH Nordstadt-Siloah Hospitals, Hannover, Germany); Smith H*, Haugstvedt AF, Jönsson ML (Bispebjerg Hospital, Denmark); Caja Vivancos P*, Villalabeitia Ateca I*, Prieto Calvo M, Marín H, Martin Playa P*, Gainza A, Aragon Achig EJ, Rodriguez Fraga A, Melchor Corcóstegui I, Mallabiabarrena Ormaechea G*, Garcia Gutierrez JJ, Barbier L, Pesántez Peralta MA*, Jiménez Jiménez M, Municio Martín JA, Gómez Suárez J, García Operé G, Pascua Gómez LA, Oñate Aguirre M (Hospital Universitario Cruces, Spain); Fernandez-Colorado A, De la Rosa-Estadella M*, Gasulla-Rodriguez A, Serrano-Martin M, Peig-Font A, Junca-Marti S, Juarez-Pomes M, Garrido-Ondono S, Blasco-Torres L, Molina-Corbacho M, Maldonado-Sotoca Y, Gasset-Teixidor A, Blasco-Moreu J (Corporació Sanitària Parc Taulí, Spain); Turrado-Rodriguez V*, Lacy AM, de Lacy FB, Morales X, Carreras-Castañer A*, Torner P, Jornet-Gibert M, Balaguer-Castro M, Renau-Cerrillo M, Camacho-Carrasco P, Vives-Barquiel M, Campuzano-Bitterling B, Gracia I, Pujol-Muncunill R (Hospital Clínic de Barcelona, Spain); Estaire Gómez M*, Padilla-Valverde D*, Sánchez-García S, Sanchez-Pelaez D, Jimenez Higuera E, Picón Rodríguez R, Fernández Camuñas À, Martínez-Pinedo C, Garcia Santos EP, Muñoz-Atienza V, Moreno Pérez A, López de la Manzanara Cano CA (Hospital General Universitario De Ciudad Real, Spain); Crego-Vita D*, Huecas-Martinez M (Hospital Central De La Defensa Gomez Ulla, Madrid, Spain; Domenech J, Roselló Añón A*, Sangüesa MJ (Hospital Arnau De Vilanova, Spain); Bernal-Sprekelsen JC*, Catalá Bauset JC, Renovell Ferrer P, Martínez Pérez C, Gil-Albarova O, Gilabert Estellés J, Aghababyan K (Consorcio Hospital General Universitario, Spain); Rivas R, Rivas F (Hospital Clinico Universitario Zaragoza, Zaragoza, Spain); Escartin J*, Blas Laina JL, Nogués A, Cros B, Talal El-Abur I, Garcia Egea J, Yanez C (Hospital Royo Villanova, Spain); Kauppila JH*, Sarjanoja E (Länsi-Pohja Central Hospital, Finland); Tzedakis S*, Bouche PA*, Gaujoux S (Hôpital

Cochin - Aphp, France); Gossot D, Seguin-Givelet A*, Fuks D, Grigoroiu M, Sanchez Salas R, Cathelineau X, Macek P, Barbé Y, Rozet F, Barret E, Mombet A, Cathala N, Brian E, Zadegan F, Conso C (Institut Mutualiste Montsouris, France); Baldwin AJ, West R*, Gammeri E, Catton A, Marinos Kouris S (Stoke Mandeville, Wycombe General, United Kingdom); Pereca J*, Singh J (University Hospital Ayr, United Kingdom); Patel P*, Handa S, Kaushal M, Kler A, Reghuram V, Tezas S (Furness General Hospital, United Kingdom); Oktseloglou V*, Mosley F*, De La Cruz Monroy MFI, Bobak P, Omar I, Ahad S, Langlands F, Brown V, Hashem M (Bradford Royal Infirmary, United Kingdom); Williams A, Ridgway A, Pournaras D, Britton E, Lostis E, Ambler GK, Chu H, Hopkins J*, Manara J, Chan M, Doe M, Moon RDC, Lawday S, Jichi T, Singleton W (Southmead Hospital, United Kingdom); Mannion R, Stewart GD*, Ramzi J, Mohan M, Singh AA, Ashcroft J, Baker OJ, Coughlin P*, Davies RJ*, Durst AZED, Abood A, Habeeb A, Hudson VE, Kolias A, Lamb B, Luke L, Mitrasinovic S, Murphy S, Ngu AWT, O'Neill JR*, Waseem S, Wong K, Georgiades F, Hutchinson PJ*, Tan XS, Pushpa-rajah J, Colquhoun A, Masterson L, Abu-Nayla I, Walker C, Balakrishnan A*, Rooney S, Irune E, Byrne MHV, Durrani A (Addenbrooke's Hospital, Cambridge University Hospitals NHS Foundation Trust, United Kingdom); Richards T*, Sethuraman Venkatesan A, Combellack T*, Williams J, Tahhan G, Mohammed M, Kornaszewska M, Valtzoglou V, Deglurkar I*, Rahman M, Von Oppell U, Mehta D, Koutentakis M, Syed Nong Chek SAH, Hill G, Morris C*, Shinkwin M, Torkington J, Cornish J (University Hospital of Wales, United Kingdom); Richards T*, Sethuraman Venkatesan A, Combellack T*, Williams J, Tahhan G, Mohammed M, Kornaszewska M, Valtzoglou V, Deglurkar I*, Rahman M, Von Oppell U, Mehta D, Koutentakis M, Syed Nong Chek SAH, Hill G, Morris C*, Shinkwin M, Torkington J, Cornish J (University Hospital of Wales, United Kingdom); Houston R, Mannan S*, Ayeni F, Tustin H, Bordenave M, Robson A (Cumberland Infirmary, United Kingdom); Vimalachandran D*, Manu N, Eardley N, Krishnan E, Serevina OL, Martin E, Smith C, Jones A, Roy Mahapatra S, Clifford R (Countess of Chester Hospital, United Kingdom); Matthews W, Mohankumar K, Khawaja I, Palepa A, Doulias T (Colchester University Hospital, United Kingdom); Premakumar Y, Jauhari Y, Koshnow Z, Bowen D, Uberai A, Hirri F, Stubbs BM* (Dorset County Hospital, United Kingdom); McDonald C*, Manickavasagam J*, Ragupathy K*, Davison S, Dalgleish S*, McGrath N, Kanitkar R, Payne CJ, Ramsay L (Ninewells Hospital, United Kingdom); Ng CE*, Collier T, Khan K*, Evans R (University Hospital North Durham, United Kingdom); Brennan C, Henshall DE, Drake T*, Harrison EM*, Zamvar V, Tambyraja A, Skipworth RJE*, Linder G, McGregor R, Brennan P, Mayes J, Ross L, Smith S, White T, Jamjoom AAB, Pasricha R (Royal Infirmary of Edinburgh, United Kingdom); Holme T, Abbott S, Razik A, Thrumurthy S, Steinke J, Baker M*, Howden D, Baxter Z, Osagie L, Bence M (Epsom & St Helier University Hospitals NHS Trust, United Kingdom); Fowler GE, Massey L, Rajaretnam N, Evans J, John J, Goubran A, Campain N, McDermott FD*, McGrath JS*, Ng M, Pascoe J, Phillips JRA, Daniels IR (Royal Devon and Exeter Hospital, United Kingdom); Raptis DA, Pollok JM*, Machairas N, Davidson B, Fusai G, Soggiu F, Xyda S, Hidalgo Salinas C, Tzerbinis H, Pissanou T, Gilliland J, Chowdhury S, Varcada M*, Hart C, Mirnezami R, Knowles J, Angamuthu N (Royal Free Hospital, United Kingdom); Vijay V*, Shakir T, Hasan R, Tansey R (Princess Alexandra Hospital, Harlow, United Kingdom); Ross E, Loubani M*, Wilkins A, Cao H, Capitelli-McMahon H, Hitchman L, Ikram H, Andronic A, Aboelkassem Ibrahim A, Totty J (Hull University Teaching Hospitals NHS Trust, United Kingdom); Tayeh S*, Chase T, Humphreys L, Ayorinde J, Ghanbari A, Cuming T (Homerton University Hospital, United Kingdom); Williams K*, Chung E, Hagger R, Karim A, Hainsworth A, Flatman M, Trompeter A, Hing C, Brown O, Tsinaslanidis P, Benjamin MW, Leyte A, Tan C, Smelt J, Vaughan P, Santhirakumaran G, Hunt I, Raza M A Labib (St George's Hospital, United Kingdom); Luo X, Sudarsanam A, Rolls A, Lyons O, Onida S, Shalhoub J, Sugand K, Park C, Sarraf KM*, Erridge S, Kinross J*, Denning M, Yalamanchili S, Abuown A, Ibrahim M, Martin G (St Mary's Hospital, United Kingdom); Davenport D, Wheatstone S* (St Thomas' Hospital, United Kingdom); Andreani SM*, Bath MF, Sahni A, Judkins N, Rigueros Springford L, Sohrabi C, Bacarese-Hamilton J, Taylor FG, Patki P*, Tanabalan C (Whipps Cross University Hospital, United Kingdom); Reynolds J, Alexander ME, Smart CJ* (Macclesfield District General Hospital, United Kingdom); Stylianides N*, Abdalla M, Newton K, Bhatia K*, Edmondson R, Abdeh L, Jones D, Zeiton M, Ismail O, Naseem H, Advani R (Manchester Royal Infirmary, United Kingdom); Fell A*, Smith A, Halkias C, Evans J, Nikolaou S, English C*, Kristinsson S, Oni T, Ilahi N, Ballantyne K, Woodward Z, Merh R (Queen Elizabeth The Queen Mother Hospital Margate, United Kingdom); Robertson-Smith B, Mahmoud A, Ameerally P, Finch JG*, Gnanachandran C, Pop I, Rogers M, Yousef Y, Mohamed I, Woods R, Zahid H, Mundy G* (Northampton General Hospital, United Kingdom); Aujayeb A, Townshend D*, McLarty N, Shenfine A, Jackson K, Johnson C (Northumbria NHS Hospital Trust, United Kingdom); Dass D, Ford D (Robert Jones and Agnes Hunt Orthopaedic Hospital, United Kingdom); Khan J*, Thiruchandran G, Toh SKC*, Ahmad Y*, Allana A, Bellis C, Babawale O, Phan YC, Lokman U, Ismail M, Koc T, Witek A, Duggleby L, Shamoon S, Stefan S, Clancy H (Queen Alexandra Hospital, United Kingdom); Singh S, Mukherjee S*, Ferguson D,

Smith C, Mansuri A, Thakrar A, Wickramarachchi L, Cuthbert R, Sivayoganathan S, Chui K, Karam E, Dott C, Shankar S (Queen's Hospital Romford, United Kingdom); Singh R*, Lane J, Colvin HV, Badran A*, Cadersa A, Williams S, Cumpstey A, Hamady Z, Aftab R, Wensley F, Byrne J, Morrison-Jones V, Sekhon GK, Shields H, Shakoor Z, Yener A, Talbot T, Khan A, Alzetani A*, Cresner R (Southampton General Hospital, United Kingdom); Johnson D*, Hughes I, Hall J, Rooney J, Chatterji S, Zhang Y, Owen R, Rudic M, Hunt J (Stepping Hill Hospital, United Kingdom); Zakai D, Thomas M*, Aladeojebi A, Ali M, Gaunt A*, Barmayehvar B, Kitchen M, Gowda M, Mansour F, Jarvis M, Halliday E, Lefroy R, Nanjaiah P, Ali S, Kitchen M* (Royal Stoke University Hospital, United Kingdom); Lin DJ, Rajgor AD, Scurrah RJ, Kang C, Watson LJ, Harris G, Royle T*, Cunningham Y, James G, Steel B, Luk ACO (Sunderland Royal Hospital, United Kingdom); Stables G*, Doorgakant A*, Thiruvasagam VG, Carter J, Reid S, Mohammed R, Marlow W (Warrington & Halton Teaching Hospitals NHS Trust, United Kingdom); Ferguson H*, Wilkin R, Konstantinou C, Yershov D, Vatish J, Denning A (South Warwickshire NHS Foundation Trust, United Kingdom); Das R* (Hampshire Hospitals NHS Foundation Trust, United Kingdom); Powell S*, Magee C*, Agarwal K*, Mangos E, Nambirajan T (Wirral University Teaching Hospital, United Kingdom); Flindall I, Mahendran V, Hanson A (Worcestershire Royal Hospital, United Kingdom); De Marchi J, Hill A*, Farrell T, Davis NF, Kearney D, Nelson T (Beaumont Hospital, Ireland); Picciariello A*, Papagni V, Altomare DF (Azienda Ospedaliero Universitaria Consorziale Policlinico Di Bari, Italy); Granieri S*, Cotsoglou C (Asst Vimercate, Italy); Cabeleira A, Branco C, Serralheiro P*, Alves R, Teles T (Hospital De Cascais - Dr. José De Almeida, Portugal); Lázaro A*, Canhoto C, Simões J, Costa M, Almeida AC, Nogueira O, Oliveira A, Athayde Nemésio R, Silva M, Lopes C, Amaral MJ, Valente da Costa A, Andrade R, Martins R,Guimarães A, Guerreiro P, Ruivo A, Camacho C, Duque M, Santos E, Breda D, Oliveira JM, De Oliveira Lopez AL, Garrido S, Colino M, De Barros J, Correia S, Rodrigues M (Centro Hospitalar E Universitário De Coimbra, Portugal); Cardoso P*, Martins R, Teixeira J, Soares AP, Morais H*, Pereira R, Revez T, Manso MI, Domingues JC, Henriques P, Ribeiro R, Ribeiro VI, Cardoso N, Sousa S, Martins dos Santos G (Centro Hospitalar Universitário do Algarve - Unidade De Faro, Portugal); Miranda P, Garrido R*, Peralta Ferreira M, Ascensão J, Costeira B, Cunha C, Rio Rodrigues L, Sousa Fernandes M, Azevedo P, Ribeiro J, Lourenço I, Gomes H, Mendinhos G*, Nobre Pinto A (Hospital Beatriz Angelo, Portugal); Cardoso P*, Martins R, Teixeira J, Soares AP, Morais H*, Pereira R, Revez T, Manso MI, Domingues JC, Henriques P, Ribeiro R, Ribeiro VI, Cardoso N, Sousa S, Martins dos Santos G (Centro Hospitalar Universitário do Algarve - Unidade De Faro, Portugal); Taflin H* (Sahlgrenska University Hospital, Sweden); Abdou H, Diaz J*, Richmond M, Clark J, O'Meara L, Hanna N (University of Maryland Medical Center, United States); Cooper Z*, Salim A*, Hirji SA, Brown A, Chung C, Hansen L, Okafor BU, Roxo V, Raut CP, Jolissaint JS, Mahvi DA (Brigham and Women's Hospital, United States); Reinke C*, Ross S, Thompson K, Manning D, Perkins, R (Atrium Health Carolinas Medical Center, United States); Volpe A, Merola, S (NewYork Presbyterian Queens, Flushing, United States); Ssentongo A, Ssentongo P, Oh JS, Hazelton J*, Maines J, Gusani N, Garner M, Horvath S (Pennsylvania State University, United States); Martin RCG*, Bhutiani N (University of Louisville Hospital and Norton Hospital, Louisville, United States); Choron R, Peck G*, Soliman F, Rehman S (Robert Wood Johnson University Hospital, United States); Abbas A, Soliman A, Kim B, Jones C, Dauer, MD E, Renza-Stingone E, Hernandez E, Gokcen E, Kropf E, Sufrin H, Hirsch H, Ross H, Engel J, Sewards J, Diaz J, Poggio J, Sanserino K, Rae L, Philp M*, Metro M, McNelis P, Petrov R, Rehman S, Pazionis T (Temple University Hospital, United States); Quintana M*, Jackson H (The George Washington University Hospital, United States) Lumenta DB*, Nischwitz SP, Richtig E, Pau M, Srekl-Filzmaier P, Eibinger N, Michelitsch B, Fediuk M, Papinutti A, Seidel G, Kahn J, Cohnert TU (Medical University of Graz, Austria); Kantor E (AP-HP Hopital Bichat Claude Bernard, Paris, France); Kahiu J*, Hossain N, Hosny S (Barnet General Hospital, United Kingdom); Sultana A*, Taggarsi M, Vitone L*, Lambert J, Vaz OP, Sarantitis I, Shrestha D, Timbrell S, Shugaba A (Royal Blackburn Hospital, United Kingdom); Jones GP, Gardner A, Tripathi SS* (Lancashire Teaching Hospitals NHS Foundation Trust, United Kingdom); Emerson H, Vejsbjerg K, Pearce L, McCormick W, Fisher A*, Singisetti K, Aawsaj Y, Barry C (Gateshead Health NHS Foundation Trust, United Kingdom); Blanco J*, Vanker R, Ghobrial M, Jones G, Kanthasamy S, Fawi H, Awadallah M, Chen F, Cheung J* (Hinchingbrooke Hospital, United Kingdom); Tingle S, Abbadessa F, Sachdeva A, Rai B*, Chan CD, McPherson I, Booth K, Mahmoud Ali F, Pandanaboyana S, Grainger T, Nandhra S, Patience A, Rogers A*, Roy C, Williams T, Dawe N, McCaffer C, Riches J, Bhattacharya S, Moir J, Kalson NS, Elamin Ahmed H, Mellor C, Saleh C, Koshy RM, Hammond J*, Sanderson L, Wahed S, Phillips AW, Ghosh K* (Newcastle Upon Tyne Hospitals NHS Foundation Trust, United Kingdom); Rogers LJ*, Labib PL, Miller D, Minto G, Hope N, Marchbank A, Emslie K, Panahi P, Ho B, Perkins C, Clough E, Roy H, Enemosah I, Campbell R, Natale J, Gohil K, Rela M, Raza N, (Derriford Hospital, United Kingdom); Menakaya C*, Webb JI, Antar M, Modi N, Sofat R, Noel J, Nunn R, Adegbola S, Eriberto F, Sharma V, Tanna R, Lodhia S (East and North Hertfordshire NHS Trust, Lister Hospital, United Kingdom); Carvalho

L, Osório C, Antunes J, Lourenço S, Balau P, Godinho M, Pereira A (Centro Hospitalar Entre o Douro e Vouga, Santa Maria da Feira, Portugal); Data Monitoring and Ethics Committee: Deborah S Keller, Neil J Smart.

**Contributors** CK is the lead author for this manuscript and was responsible for inception, analysis and writing of this manuscript, working in a team alongside the COVIDSurg Collaborative. AM is the the overall guarantor for this study. A full list of authors and roles within this research project is listed.

**Funding** This work was supported by a National Institute for Health Research (NIHR) Global Health Research Unit Grant (NIHR 16.136.79), the Association of Coloproctology of Great Britain and Ireland, Association of Upper Gastrointestinal Surgeons, Bowel Disease Research Foundation, Yorkshire Cancer Research, Sarcoma UK, the British Association of Surgical Oncology, the Vascular Society for Great Britain and Ireland and the European Society of Coloproctology.

**Disclaimer** The funders had no role in study design, data collection, analysis and interpretation or writing of this report. The views expressed are those of the authors and not necessarily those of the National Health Service, the NIHR, or the UK Department of Health and Social Care.

**Competing interests** None declared.

**Patient consent for publication** Not applicable.

**Ethics approval** The study protocol was registered online (ClinicalTrials.gov identifier: NCT04323644).

**Provenance and peer review** Not commissioned; externally peer reviewed.

**Data availability statement** Data are available upon reasonable request. All data relevant to the study are included in the article or uploaded as supplementary information.

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
