## [Reviewer comments · BMJ Open]

ARTICLE DETAILS

TITLE (PROVISIONAL)	Outcomes after peri-operative SARS-CoV-2 infection in patients with proximal femoral fractures: an international cohort study
AUTHORS	Khatri, Chetan

VERSION 1 – REVIEW

REVIEWER	Martin Sigurdsson Landspítali University Hospital
REVIEW RETURNED	14-Mar-2021

GENERAL COMMENTS	Thank you for the opportunity to review this study from the multicenter COVIDSurg consortium on surgical repair of femoral fractures. This consortium will provide us with valuable information, and given the demographics of femoral fractures, it can certainly be expected that a recent or coexisting COVID19 infection results in worse outcomes. Overall, I think the manuscript is of a high quality, the methodology and reporting of results are sound and the authors highlight their limitations appropriately. I have the following suggestions that I hope will improve the manuscript. Major points 1. My big question is – how the authors propose their results are used to further the care of patients with COVID19 who undergo acute orthopedic surgery – since delaying the surgical care to further recovery from COVID19 is usually not an option? A lot of the discussion is written in the sense that the clinician and the patient know at the time of surgery if the patient will contract SARS-CoV2 infection in the postoperative period. The authors imply this could even tilt the balance for undergoing surgical management vs. noninvasive care (with a very high mortality) towards noninvasive care, and this could affect surgical decisions. However, since the additional risk is contracting SARS-CoV2 postoperatively, this is not at the time of surgery, and therefore this argument is invalid.2. I would also highly recommend that the authors refrain from interpreting the “absence of evidence” as the “evidence of absence” in their discussion chapter. Not finding an association between choice of anesthetic and outcome does mean there is not one, and this is certainly not a randomized choice. Similarly, most individuals managing patients with COVID19 would suggest a period of at least 14 days is needed to recover from an infection. Given that the study only correlated a delay of up to 3 days with no increased mortality, it is impossible to claim that it might be appropriate to delay the case for medical optimization for COVID19, as the duration of the delay tested is relatively short.
---

	3. I would advise restructuring of the results chapter. I might suggest moving the subchapter “Diagnosis” to the beginning of the results chapter (as it is really a patient demographic) followed by preoperative variables and procedure description. and move “Mortality” to the end as it is the primary outcome. This could then include the association with pre-operative variables and time of SARS-CoV2 diagnosis. 4. Given the very large number of tests perform, the authors need to justify their selection of a significant p-value 5. Is it possible that the portion of patients diagnosed pre-procedurally were actually more likely to have recovered from their COVID19 disease, while those diagnosed post-operative were more likely to have active disease? For example, are their symptom descriptions different? 6. I certainly agree that 29.4% 30-day mortality is high, but the authors report that they have a high-quality cohort with much less frequency of this outcome, and that this compares unfavorably. However, as noted by the authors there is no control group, so the authors need to convince the reader that the historical one-nation cohort is at least comparable to the subset of patients from UK or better yet the entire cohort. Are the patient and procedural characteristics similar at least? There is an obvious selection bias since the group who contracts COVID19 differs from all-comers who have a femoral fracture. I wonder what the comparison would be if they only compared the outcomes with a group of patients who had infectious or pulmonary complications in non-COVID19 era? Minor issues 1. Page 3 line 38 – a p-value of 0.000 is impossible, please reformat to <0.001 or provide an exact number. This occurs at more instances in the manuscript. 2. Page 3 line 55 – This is a style point but generally I would not start a sentence with a number. This occurs at more instances in the manuscript. 3. Page 4 Line 7 – Also a style point, but given the vast amt of literature, I would refrain from caliming “first to” or “largest”. Does not add much value, and can easily become inaccurate 4. Page 8 line 159 – I think the inclusion criteria needs to be outlined differently. The reader might assume that all patients hat RT-PCR. You need to describe this as either clinical diagnosis, radiological diagnosis or by a positive SARS-CoV-2 5. Page 10 line 208 – I do not think any patient in this study could classify as ASA V (immediate loss of limb or life if not performed within 24 hours) – can you confirm that this is the case 6. Page 13 – line 268 – I don’t understand this sentence “Chronic Obstructive Pulmonary Disorder (COPD) showed a signal to be a risk factor, however was not significantly associated, (OR 1.42, 95% CI 0.96-2.09, p=0.076)” – what does a signal to be a risk factor mean. Additionally “risk factor” implies causality and is better avoided. 7. Page 14 – Why do the authors propose there was a difference in mortality in march but compared with both February and April? 8. Tables – I would use “died” and not “dead”
--	---

REVIEWER	Marco Zuin University of Ferrara, Department of Translational Medicine
REVIEW RETURNED	02-Jul-2021

GENERAL COMMENTS	The manuscript analysed the 30-day mortality associated with peri-operative infection of patients undergoing surgery for proximal femoral fractures and examined the actors that influence mortality in a multi-variate analysis. Overall, the manuscript is well-written and timely topic. I have only some minor suggestions to the authors:  - As known, COVID-19 patients have a higher risk and incidence of venous thromboembolism, which per se, I also related to femoral fracture (Roncon L, Zuin M, Barco S, Valerio L, Zuliani G, Zonzin P, Konstantinides SV. Incidence of acute pulmonary embolism in COVID-19 patients: Systematic review and meta-analysis. Eur J Intern Med. 2020 Dec;82:29-37. doi: 10.1016/j.ejim.2020.09.006. Epub 2020 Sep 17. PMID: 32958372; PMCID: PMC7498252). I suggest to the authors to report how many patients developed this complication which is common to both disease, also including the rate of deep vein thrombosis. Obviously, this data cannot be accurate, since generally only symptomatic patients or those presenting worsening symptoms are generally evaluated with computed tomography angiography. - At the same manner, it would be useful to the reader have some data regarding the thromboprophylaxis used in these patients, which may have influenced the study outcome. - How many patients required ICU admission and was this clinical setting associated with a worst outcome?
---

VERSION 1 – AUTHOR RESPONSE

Reviewer 1

Major points	
My big question is – how the authors propose their results are used to further the care of patients with COVID19 who undergo acute orthopedic surgery – since delaying the surgical care to further recovery from COVID19 is usually not an option? A lot of the discussion is written in the sense that the clinician and the patient know at the time of surgery if the patient will contract SARS-CoV2 infection in the postoperative period. The authors imply this could even tilt the balance for undergoing surgical management vs. noninvasive care (with a very high mortality) towards noninvasive care, and this could affect surgical decisions. However, since the additional risk is contracting SARS-CoV2 postoperatively, this is not at the time of surgery, and therefore this argument is invalid.	Thank you for your comments. We have clarified the discussion section to distinguish how these findings should be used by clinicians. We have explicitly structured our discussion about the management of patients with and without a diagnosis of SARS-CoV-2 at the time of presentation. For those with a positive diagnosis, we have discussed the role of this data informing consent. As such, with higher mortality rates, to therefore discuss with the patient and/or family with the risks of operative vs non-operative care.

	For the patient who presents without SARS-CoV-2 infection we have highlighted a suggestion that majority of transmission occurs in the health care setting. As such we have given examples of methods to reduce spread in hospital, and further reinforced messages to reduce transmission in the community.
I would also highly recommend that the authors refrain from interpreting the “absence of evidence” as the “evidence of absence” in their discussion chapter. Not finding an association between choice of anesthetic and outcome does mean there is not one, and this is certainly not a randomized choice. Similarly, most individuals managing patients with COVID19 would suggest a period of at least 14 days is needed to recover from an infection. Given that the study only correlated a delay of up to 3 days with no increased mortality, it is impossible to claim that it might be appropriate to delay the case for medical optimization for COVID19, as the duration of the delay tested is relatively short.	Thank you, we have changed our discussion to place the choice of anaesthetic as a suggestion, highlighting that this was not the primary outcome in this study. In addition, we have highlighted this is an exploratory study and therefore this data should be interpreted with caution. We have further clarified timescales and examples for medical optimisation suggested in our discussion. As correctly identified, three days would not make a significant reduction in risk of mortality from SARS-CoV-2 infection. Given that many clinical guidelines recommend surgery within 36 hours for people with proximal femoral fractures, we do believe this is a useful finding giving a short window for optimisation, although we recognise the same weaknesses of this finding as for the choice of anaesthetic, and it is prone to bias. Medical optimisation of other co-morbidities commonly found in this population, including issues such as concurrent acute renal failure, electrolyte disturbance and/or anticoagulation related issues can also be considered.
I would advise restructuring of the results chapter. I might suggest moving the subchapter “Diagnosis” to the beginning of the results chapter (as it is really a patient demographic) followed by preoperative variables and procedure description. and move “Mortality” to the end as it is the primary outcome. This could then include the association with pre-operative variables and time of SARS-CoV2 diagnosis.	Thank you, we feel our methods section has been appropriately structured and would request we continue to keep it as it has been presented. We present basic demographics, and then followed by the primary outcome measure-mortality. This includes our adjusted risk factors which is the highlight of this paper. Next is present the secondary outcome measure by means of pulmonary complications.

	We present other subchapters such as diagnosis, preoperative variables and procedure descriptions as we report the mortality associated with these factors. However, as these are not adjusted in the logistic regression model, we feel it would be misleading to present this before our adjusted analysis. As correctly identified by the reviewer (and corrected by ourselves), the absence of an association between variables and mortality in a non-adjusted analysis should be interpreted with caution. Resultantly, we have made this clear to prevent misleading readers.
Given the very large number of tests perform, the authors need to justify their selection of a significant p-value	Thank you, this was an exploratory analysis where we tested at the 5% significance level. As it was not strictly a test of an intervention or a confirmatory analysis, no specific adjustments were made for model testing. We have clarified this and included it in the statistical analysis of the methods section.
Is it possible that the portion of patients diagnosed pre-procedurally were actually more likely to have recovered from their COVID19 disease, while those diagnosed post-operative were more likely to have active disease? For example, are their symptom descriptions different?	Thank you, please note the inclusion criteria for pre-operative diagnosis is 7 days rather than post-operative which extends to 30 days. This is following the virology and pathogenesis of SARS-CoV-2 where 7 days represents the peak of mild to moderate infection, and the start of severe infection. https://www.bmj.com/content/371/bmj.m3862). This 7-day period was chosen to reflect that people with infection would be likely to have symptomatic disease at the time of procedure and not have recovered from it. We have further clarified this within the methods of this manuscript.

I certainly agree that 29.4% 30-day mortality is high, but the authors report that they have a high-quality cohort with much less frequency of this outcome, and that this compares unfavorably. However, as noted by the authors there is no control group, so the authors need to convince the reader that the historical one-nation cohort is at least comparable to the subset of patients from UK or better yet the entire cohort. Are the patient and procedural characteristics similar at least? There is an obvious selection bias since the group who contracts COVID19 differs from all-comers who have a femoral fracture. I wonder what the comparison would be if they only compared the outcomes with a group of patients who had infectious or pulmonary complications in non-COVID19 era?	We have clarified this mortality range to include data from multiple, high-quality national registries and systematic reviews from a broad geographical spread of different health care settings which all demonstrate a 30-day mortality of less than 10%. We hope that this should reinforce that no matter the population or setting, mortality greater than 10% at 30-days should be seen to be an outlier to current standard of practice.
Minor points	
Page 3 line 38 – a p-value of 0.000 is impossible, please reformat to <0.001 or provide an exact number. This occurs at more instances in the manuscript.	Thank you, this has been corrected.
Page 3 line 55 – This is a style point but generally I would not start a sentence with a number. This occurs at more instances in the manuscript.	Thank you, this has been changed where appropriate.
Page 4 Line 7 – Also a style point, but given the vast amt of literature, I would refrain from caliming “first to” or “largest”. Does not add much value, and can easily become inaccurate	This has been edited.
Page 8 line 159 – I think the inclusion criteria needs to be outlined differently. The reader might assume that all patients hat RT-PCR. You need to describe this as either clinical diagnosis, radiological diagnosis or by a positive SARS-CoV-2	This has been clarified. The subsequent paragraph details criteria for each diagnosis of laboratory, clinical or radiological diagnosis.
Page 10 line 208 – I do not think any patient in this study could classify as ASA V (immediate loss of limb or life if not performed within 24 hours) – can you confirm that this is the case	Thank you, we can confirm no patient entered into this study was ASA V.
Page 13 – line 268 – I don’t understand this sentence “Chronic Obstructive Pulmonary	This has been changed to say COPD showed

Disorder (COPD) showed a signal to be a risk factor, however was not significantly associated, (OR 1.42, 95% CI 0.96-2.09, p=0.076)” – what does a signal to be a risk factor mean. Additionally “risk factor” implies causality and is better avoided.	no significant association.
Page 14 – Why do the authors propose there was a difference in mortality in march but compared with both February and April?	Thank you, we have added a paragraph in our discussion linking the caseload of SARS-CoV-2 to increased mortality.
Tables – I would use “died” and not “dead”	This has been changed

Reviewer 2

As known, COVID-19 patients have a higher risk and incidence of venous thromboembolism, which per se, I also related to femoral fracture (Roncon L, Zuin M, Barco S, Valerio L, Zuliani G, Zoncin P, Konstantinides SV. Incidence of acute pulmonary embolism in COVID-19 patients: Systematic review and meta-analysis. Eur J Intern Med. 2020 Dec;82:29-37. doi: 10.1016/j.ejim.2020.09.006. Epub 2020 Sep 17. PMID: 32958372; PMCID: PMC7498252). I suggest to the authors to report how many patients developed this complication which is common to both disease, also including the rate of deep vein thrombosis. Obviously, this data cannot be accurate, since generally only symptomatic patients or those presenting worsening symptoms are generally evaluated with computed tomography angiography.	We agree that the question of the incidence and subsequent management of venothromboembolism (VTE) within this population is important. As correctly identified, people suffering from lower limb fractures, particularly femoral fractures are susceptible. This was outside of the scope of this study and therefore, we did not collect this data. Due to the combined methodological and clinical difficulty in accurately diagnosing VTE (as highlighted by the reviewer), to gain meaningful conclusions, this would require a dedicated study with focus on VTE. As such, the COVIDSurg group is currently planning to collect data specifically looking at VTE within SARS-CoV-2 positive patients. We hope to be able to share the results of this in due course.
At the same manner, it would be useful to the reader have some data regarding the thromboprophylaxis used in these patients, which may have influenced the study outcome.	As stated above, this was outside of the scope of this study and data on thromboprophylaxis was not collected within this study.
How many patients required ICU admission and was this clinical setting associated with a worst outcome?	Unfortunately, we did not collect data on the location of treatment for people in this study and as such are unable to report this.

	There is an inference that a significant majority of people suffering from acute respiratory distress syndrome (ARDS), which is a part of the pulmonary complications described in this study, are treated in intensive care units. Mortality and factors that influence it within the intensive care environment are described in detail by the intensive care national audit and research centre (ICNARC) reports (https://www.icnarc.org/our-audit/audits/cmp/reports). As such high-quality reports are produced from this group, we felt unnecessary to repeat this.
--	---

VERSION 2 – REVIEW

REVIEWER	Martin Sigurdsson Landspítali University Hospital
REVIEW RETURNED	25-Aug-2021
GENERAL COMMENTS	Authors have responded to feedback adequately by my opinion